# Structure/Activity Analysis of TASK-3 Channel Antagonists Based on a 5,6,7,8 tetrahydropyrido[4,3-d]pyrimidine

**DOI:** 10.3390/ijms20092252

**Published:** 2019-05-07

**Authors:** David Ramírez, Mauricio Bedoya, Aytug K. Kiper, Susanne Rinné, Samuel Morales-Navarro, Erix W. Hernández-Rodríguez, Francisco V. Sepúlveda, Niels Decher, Wendy González

**Affiliations:** 1Instituto de Ciencias Biomédicas, Facultad de Ciencias de la Salud, Universidad Autónoma de Chile. El Llano Subercaseaux 2801-Piso 6, 7500912 Santiago, Chile; 2Centro de Bioinformática y Simulación Molecular (CBSM), Universidad de Talca. 1 Poniente No. 1141, 3460000 Talca, Chile; maurobedoyat@gmail.com (M.B.); erhernandez@utalca.cl (E.W.H.-R.); 3Institute for Physiology and Pathophysiology, Vegetative Physiology, Philipps-University of Marburg, Deutschhausstraße 2, 35037 Marburg, Germany; aytug.kiper@staff.uni-marburg.de (A.K.K.); rinne@staff.uni-marburg.de (S.R.); 4Bachillerato en Ciencias, Facultad de Ciencias, Universidad Santo Tomás, Av. Circunvalación Poniente #1855, 3460000 Talca, Chile; smoralesn@santotomas.cl; 5Escuela de Química y Farmacia. Facultad de Medicina. Universidad Católica del Maule, 3460000 Talca, Chile; 6Centro de Estudios Científicos (CECs), Arturo Prat 514, 5110000 Valdivia, Chile; fsepulveda@cecs.cl; 7Millennium Nucleus of Ion Channels-Associated Diseases (MiNICAD), Universidad de Talca, 3460000 Talca, Chile

**Keywords:** TASK-3 channel, 5,6,7,8 tetrahydropyrido[4,3-d]pyrimidine derivatives, PK-THPP, TASK channels blockers, mutagenesis screen, molecular docking, molecular dynamics, drug-protein interaction

## Abstract

TASK-3 potassium (K^+^) channels are highly expressed in the central nervous system, regulating the membrane potential of excitable cells. TASK-3 is involved in neurotransmitter action and has been identified as an oncogenic K^+^ channel. For this reason, the understanding of the action mechanism of pharmacological modulators of these channels is essential to obtain new therapeutic strategies. In this study we describe the binding mode of the potent antagonist PK-THPP into the TASK-3 channel. PK-THPP blocks TASK-1, the closest relative channel of TASK-3, with almost nine-times less potency. Our results confirm that the binding is influenced by the fenestrations state of TASK-3 channels and occurs when they are open. The binding is mainly governed by hydrophobic contacts between the blocker and the residues of the binding site. These interactions occur not only for PK-THPP, but also for the antagonist series based on 5,6,7,8 tetrahydropyrido[4,3-d]pyrimidine scaffold (THPP series). However, the marked difference in the potency of THPP series compounds such as 20b, 21, 22 and 23 (PK-THPP) respect to compounds such as 17b, inhibiting TASK-3 channels in the micromolar range is due to the presence of a hydrogen bond acceptor group that can establish interactions with the threonines of the selectivity filter.

## 1. Introduction

Two-pore domain potassium (K_2P_) channels have been widely studied since the *KCNK* gene family (encoding these proteins) was discovered [1], providing important advances in the understanding of their physiological roles. The TASK (TWIK-related acid-sensitive K^+^) channel subfamily includes three members (TASK-1, -3 and -5) [2]. The closest relative of the TASK-3 channel [3] is TASK-1 [4], with a sequence identity of ca. 58.9% determined between the human variants [5]. TASK-3 plays an important role under physiological conditions and is very sensitive to extracellular pH changes in the range of 6 to 7 [3,6,7].

The tertiary structure of K_2P_ channels is unique in relation to other potassium channels. The crystallized structures of the K_2P_ channels TWIK-1 (PDB: 3UKM [8]), TRAAK (PDBs: 3UM7 [5], and 4I9W [9]), TREK-2 (PDBs: 4BW5, 4XDJ, 4XDK and 4DKL [10]) and TREK-1 (PDBs: 4TWK, 6CQ6 and 6CQ8 [11]) reveal differences that give structural insights into distinctive gating and ion permeation properties. Near to the center of the membrane, the M2 transmembrane segment is kinked by approximately 20°, generating two lateral cavities (fenestrations) that connect the inner pore with the membrane [12]. These fenestrations have an essential role in the modulation of K_2P_ channels [13,14] acting as binding pockets for drugs like norfluoxetine, the active metabolite of Prozac^®^, [10] or BL1249 [15] in TREK-2.

Not many promising high-potency TASK-3 inhibitory modulators have been identified so far. The first potent TASK-3 blocker was reported in 2012 by Merck et al. [16]. They synthetized a series of derivatives based on 5,6,7,8-tetrahydropyrido [4,3-d] pyrimidine scaffold (THPP series), where the compound PK-THPP (IC_50_ = 35 nM) exhibits the highest inhibitory effect on TASK-3 using a voltage sensitive fluorescent dye approach (FLIPR assay) and an IonWorks Quattro electrophysiology assay for IC_50_ measurement. Then, Flaherty et al. [17] reported the application of bis-amide derivatives as novel TASK modulators, where the most potent and selective compound exhibits an IC_50_ = 16 nM for TASK-1 with 62-fold selectivity over TASK-3 in QPatch automated electrophysiology assay. The most potent compound against TASK-3 reported by Flaherty et al. presents an IC_50_ = 38 nM.

Moreover, the binding mode of only a few TASK blockers and other K_2P_ channels blockers is well known. Using a functional mutagenesis approach and molecular simulations, our group has studied the binding mode of the blocker A1899 [18] and other inhibitory compounds [19] of TASK-1 channels, suggesting an intracellular TASK channel pore binding site where the fenestrations might provide a physical anchor, reflecting an energetically favorable binding mode that, after pore occlusion, stabilizes the closed state of the channels [13] (Figure 1A). Recently, we showed that the local anesthetic bupivacaine blocks TASK-1 laterally, in the side fenestrations [14] (Figure 1B). This allosteric interaction was described for the TREK-2 channel blocker norfluoxetine [10] (Figure 1C) and recently for the activator BL1249 [15]. The PK-THPP binding site was previously explored by Chokshi et al. in TASK-3, who identified L122, L239 and G236 as key residues because IC_50_ of PK-THPP in L122D, G236D and L239D mutants increased to >10 µM, 7 µM, and 895 nM, respectively (PK-THPP IC_50_ in WT was 10 nM). Aspartate scanning mutagenesis also suggested that residue V242 is part of the drug binding site (PK-THPP IC_50_ in TASK3-V242D was about 1.6 µM) [20]. We consider that the introduction of negative charged residues such as aspartate might dramatically disrupt the environment of TASK-3 druggable cavities, changing structures and conformation along with the side chain physical-chemical properties, thereby complicating analyses of results.

Here, we perform a structure/activity analysis of TASK-3 channel antagonists of the THPP series; this analysis type has been successfully used to understand the binding mode of a given set of ligands [21,22,23]. We started with an alanine mutagenesis screening of PK-THPP binding site. Our aim was to understand how small changes in the blocker structure significantly influence the binding affinity against TASK-3. Our results confirm that the binding is influenced by the fenestrations and occurs when they are open. The binding is mainly governed by hydrophobic contacts between the blockers and residues of the binding site mainly exposed to the pore, but some of them occur at the interface between the fenestration and the central cavity (Figure 1D). However, the marked difference in the potency of compounds of the THPP series, such as 20b, 21, 22 and 23 (PK-THPP), in relation to compounds such as 17b, inhibiting TASK-3 channels in the micromolar range, is due to the presence of a hydrogen bond acceptor group that can establish interactions with the threonines of the selectivity filter.

We consider that our exploration of the bound blocker conformation for antagonists of THPP series sheds light on the molecular basis of TASK channels inhibition, taking into account that compounds of THPP series, such as 21 to 23, show a more potent inhibitory effect for TASK-3 than for TASK-1 [16]. This is a unique feature because, excepting the compounds of THPP series, the other known TASK channel blockers are more potent for TASK-1 channels [17,18].

## 2. Results

### 2.1. Exploring the PK-THPP Binding Site in TASK-3 by Alanine Mutagenesis Screening

Chokshi et al. [20] reported that TASK-3 mutants L122D, G236D, L239D, and V242D are resistant to being blocked by PK-THPP. However, we consider that introducing negative moieties such as aspartate might affect the amino acidic environment of the PK-THPP binding site, providing inaccurate results.

To explore the PK-THPP (Figure 2A) binding site, we first tested its affinity for TASK-3 channels expressed in Xenopus oocytes by two-electrode voltage clamp (TEVC). PK-THPP blocks TASK-3 channels with an IC_50_ of 243 ± 23.8 nM (Figure 2B). The reported IC_50_ of PK-THPP for human TASK-3 channels expressed in the cell line HEK293 measured by an IonWorks Quattro electrophysiology assay was 35 nM [16], which is consistent with the notion that IC_50_ values determined in oocytes are higher than those determined in mammalian cells [18]. Following this, we performed an alanine-mutagenesis screening of the pore-lining residues of the M2 and M4 segments, as well as the threonines at the bottom of the selectivity filter (Figure 2C). We mutated the screened residues by alanine to delete the side chain of the studied amino acids and check how the affinity was affected.

The alanine-screening revealed that residues L122, L239 and G236 are essential for PK-THPP inhibition of TASK-3 channels but not the residue V242, as Chokshi found [20]. Besides, new residues were determined as part of the PK-THPP binding site, such as Q126, G231, A237, L244, L247 and T248. When the A1899 binding site [18] and binding mode [13] were described in TASK-1, some of these residues (L122, G236, L239 and L247, which is M247 in TASK-1) were also found to be essential for drug-binding, which seems reasonable, as the two molecules (PK-THPP and A1899) exhibit structural similarities, especially in the hydrophobic moieties.

Based on the alanine-screening results, we homology-modeled the TASK-3 channel structure and docked into a grid box PK-THPP and the other compounds of the THPP series. The grid box was centered in the residues L122 and L239 because their alanine mutants reduce the percentage of inhibition at least twice. The different configurations of TASK-3 models allowed us to identify the residues L122, L239 and other residues of the binding site facing the pore but also the fenestrations (Figure 2D).

### 2.2. TASK-3 Modeling and Structural Characterization

Since the three-dimensional (3D) structure of TASK-3 has not been solved, three homology models were built using TREK-1, TREK-2, and TWIK-1 as templates (see Materials and Methods section). These structures have differences in fenestration states (they could be open or closed); therefore, the different TASK-3 modeled present the same differences in the fenestration cavities. TASK-3 sequence shares 26.2% identity with TREK-2, 26.2% with TREK-1 and 27.2% with TWIK-1 [5]; therefore, they are in an acceptable range of sequence identity to be used as templates to build comparative models [24]. The TASK-3 models were subjected to 25 ns molecular dynamics simulation (MDs). The root mean squared deviations (RMSDs) of the TASK-3 backbone atoms as a function of the simulation time using their initial configuration as reference are shown in Appendix A; the dependencies of the RMSD values were tested to check whether the MDs trajectories were stable. RMSD values show that the models are stable in the last 10 ns, fluctuating less than 0.3 Å. To analyze the pore differences in TASK-3 models with the fenestration open (T3twiOO, T3tre2OO) and closed (T3tre1CC), HOLE radius profiles were calculated (Figure 3A); the profiles reveal that the TASK-3 up-state (closed fenestrations) presents a bigger pore diameter at the central cavity than the down-state (open fenestration) configuration. The models were examined and it was confirmed that the state of the fenestrations (close or open) was maintained during the trajectories (Figure 3B).

For each model we examined the relative presence of hits (Figure 2C,D) during the last 10 ns of MDs in the central cavity and/or fenestrations using the HOLE algorithm. Appendix A summarizes whether the residues of the THPP binding site face into the fenestration and/or the pore. Q126, L244, L247 and T248 are exclusively present in the pore of the channel during the MDs. Residues L122, G236 and L239 are present both in the central cavity and in the fenestrations; and G231, A237 are buried into the protein.

### 2.3. Structural-Activity Relationship (SAR) of THPP Series against TASK-3 Channel

Twenty-nine compounds of THPP series with IC_50_ < 100 µM (Appendix A) were docked against the TASK-3 models at the last frame of the 25ns MDs; then, the MM-GBSA ΔG_bind_ was calculated to correlate the THPP series structural binding mode and their reported activity against TASK3 [16]. The calculated relative ΔG_bind_ values against pIC_50_ (Appendix A) are plotted in Figure 4A. The implemented computational protocol (docking + MM-GBSA) has been successfully used in SAR studies of a given compound series against specific targets [21,25,26,27]. A low correlation coefficient was obtained for the study done against the T3tre1CC model (*R*^2^ = 0.491) and moderate correlation coefficients (T3twiOO: *R*^2^ = 0.687 and T3tre2OO: *R*^2^ = 0.753) were obtained against TASK-3 models with open fenestrations. The best SAR correlation was obtained for the T3tre2OO model, where the relative ΔG_bind_ energy values of THPP analogues varied between −84.1 (for PK-THPP) to −59.2 kcal/mol (for compound 17b).

To understand how small changes in the blocker structure significantly influence the binding affinity against TASK-3, we selected five compounds of the THPP series (Figure 4B). Compounds 20b, 21 and 22 exhibit a similar biological activity to PK-THPP, and compound 17b presents around a 1000-fold lower affinity than PK-THPP (Table 1). We selected the best docking poses for these compounds obtained in the T3tre2OO model because the highest structure-activity correlation (Figure 4A) between the THPP series and TASK-3 was found in this model. Compounds PK-THPP, 21, 20b and 22 present a similar biological activity due to their high structural similarity, with IC_50_ between 35 and 70 nM (Figure 4B), as well as a similar relative free energy, with relative ΔG_Bind_ between −84.1 and −80.5 kcal/mol (Table 2). All of them have the same THPP scaffold and a substituted piperidine (Figure 4B), and their biphenyl moiety present lipophilic interactions at the bottom of the inner cavity (Appendix A). Compound 17b has an IC_50_ = 27 µM and relative ΔG_Bind_ = −59.2 kcal/mol. This decreased affinity in comparison with the other analyzed compounds (Table 2) is due to the unsubstituted pyrrolidine ring (Figure 4B). This short moiety makes it impossible for this compound to establish interactions in the upper side of the fenestration’s inner cavity (Appendix A). The major difference in the MM-GBSA ΔG_Bind_ for all THPP analyzed compounds lies in the ΔE_vdW_ term, where compound 17b presents a significant difference of 17.9 kcal/mol in comparison with PK-THPP; the non-polar contribution (ΔG_SA_) also presents a 11.6 kcal/mol difference in comparison to PK-THPP (Table 2).

Although hydrophobic interactions with hits (Table 2 and Appendix A) govern the binding of compounds of THPP series, the differences in their biological activities—where compound 17b presents around 1000-fold lower affinity than the rest of the studied compounds—is related to the unsubstituted pyrrolidine ring in 17b versus the substituted piperidine group of the other compounds. To better understand the effect of both groups in the binding, we used the compound’s molecular features to define a common pharmacophore. Compounds 20b, 21, 22 and PK-THPP (23) share the pharmacophore (Figure 4B) with three aromatic rings (*R*), two hydrophobic groups (*H*) and two hydrogen bond acceptor groups (*A*). One of the *A* groups is not present in the 17b compound and, in consequence, it cannot accept hydrogen bonds from threonines of the selectivity filter (e.g., T93 and T199), which is an essential interaction of the potent blockers of TASK channels such as compounds A1899 [13,18], 20b, 21 (Appendix A), 22 and PK-THPP.

The biphenyl group of compounds of the THPP series has the main contribution to hydrophobic interactions with the hits (Table 2). However, the pyrimidine ring has the major contribution to the negative relative ΔG_Bind_ of the compounds (Appendix A). The pyrimidine ring establishes hydrophobic interactions with L239 in the analyzed compounds of THPP series, except in compound 22, where a hydrogen bond connects with Q126 (Appendix A). When compounds such as 21 (Appendix A) are able to establish hydrogen bonds with threonines of the selectivity filter, the major contribution to the negative relative ΔG_Bind_ is in the *A* pharmacophoric feature (Figure 4B), missing in compound 17b. In fact, the unsubstituted pyrrolidine of compound 17b does not contribute to the negative relative ΔG_Bind_, but the substituted piperidine of high affinity compounds 20b to 23 does, or at least (as in compound 20b in the current pose), does not establish unfavorable interactions (Appendix A).

### 2.4. Defining the PK-THPP Binding Mode Using a Wide Conformational Sampling

Chokshi et al. [20] showed that PK-THPP binds asymmetrically along one wall of the pore and proposed two poses for this molecule. In one pose, the biphenyl group extends pointing toward the selectivity filter and in the other, the biphenyl folds and crosses the pore. However, in our docking and MM-GBSA studies of PK-THPP, and the other four compounds of the series where the binding was reviewed in detail (Figure 4B), the biphenyl moiety points toward the bottom of the inner cavity (Appendix A). Besides, the proposed H-bonds between the compounds of THPP series with threonines of the selectivity filter could not be appreciated in the docking pose of PK-THPP selected by the SAR relationship.

A wide conformational sampling by massive docking simulations was performed to study the binding of PK-THPP with the TASK-3 channel using different conformations of the target. To consider the flexibility of the residues in TASK-3 binding site, we selected different conformations (one snapshot was taken each 1 ns) along the last 10 ns of all MDs of TASK-3 models (Appendix A). Thus, 100 different docking poses for each model (300 in total) were selected and clustered using an RMSD matrix (see Material and Methods section). In total, 40 clusters were obtained (Appendix A). Statistically significant clusters, for which the populations depart by more than two standard deviations (SD) from the mean cluster population [28,29], are summarized in Table 3. The RMSD matrices before and after clustering of PK-THPP docking solutions are shown in Appendix A, illustrating the significant clusters and their size as blue squares, visible on the diagonal lines. From the clustering process, it can be seen that PK-THPP poses docked in the T3tre2OO model exhibit lower RMSDs than the other two models. More blue shades appear in the diagram the lower the RMSD. In T3twiOO and T3tre2OO, the significant clusters interact with the interface between the fenestrations and the central cavity (Figure 5).

All PK-THPP poses were rescored by their MM-GBSA ΔG_bind_ (kcal/mol). In models with open fenestrations, the PK-THPP docking pose with the best relation between the relative ΔG_Bind_ per model and its experimental biological activity (pIC_50_) (Figure 4A, PK-THPP poses in T3twiOO and T3tre2OO models) is in the most populated clusters. For T3twiOO, pose no.84 (located in cluster no. 10) exhibits a relative ΔG_bind_ = −115.8 kcal/mol (Table 4). For T3tre2OO, pose 177 (located in cluster no. 23) exhibits a relative ΔG_bind_ = −84.1 kcal/mol (Table 1 and Table 4). However, the PK-THPP docking pose with the best relation between the relative ΔG_Bind_ and their pIC_50_ (pose no. 270) interacting with model T3tre1CC is not in a significant cluster (Table 4). We hypothesize that TASK-3 down-state, with open hydrophobic fenestrations (Figure 3) as well as narrowest pore diameter (Figure 3B), will allow PK-THPP to interact with a higher affinity, favoring lipophilic interactions in the pore. This hypothesis was tested by analyzing the energetic terms in each PK-THPP–TASK-3 model complex, as indicated in Table 4. The lipophilic term (ΔG_SA_) corresponding to the non-polar contribution to the MM-GBSA ΔG_bind_ is lower when TASK-3 is in the down-state (T3twiOO and T3tre2OO models) compared to the up-state (T3tre1CC model). We also notice that the up-state complex presents the highest electrostatic solvation energy (ΔG_GB_ = 34.6 kcal/mol). This allows us to conclude that PK-THPP interacts preferentially with structures with open fenestrations in the down-state, where the inner cavity has a smaller diameter (Figure 3B). We found similar results when characterizing A1899 binding mode with TASK1, where the open fenestrations allow A1899 to anchor inside [13].

### 2.5. Interaction of PK-THPP with Residues of TASK-3 Binding Site by Molecular Dynamics Simulations (MDs)

The PK-THPP pose 177 (Appendix A) obtained by SAR correlation in the T3tre2OO model was selected to analyze the ligand–protein interactions. This PK-THPP pose is also in the most populated cluster found in T3tre2OO model. We studied how the channel and ligand atoms interact (Figure 6A) and change (Appendix A) during an unrestrained 250 ns simulation. By monitoring the RMSD, it could be seen that PK-THPP remained anchored to its initial position at the binding site and the TASK-3 structure remained stable over the MDs, although the structures showed minor deviations from their initial positions (Appendix A). Throughout the trajectory the fenestrations remain open, mainly the left fenestration (Appendix A). However, without the drug in the binding site, the T3tre2OO model converts into the up state, resulting in a closure of the side fenestrations (Appendix A). When the left fenestration is closed, residues I118 (chain B) and L239 (chain A) can establish hydrophobic interactions. In the MDs of the TASK-3 model with PK-THPP, the butanone tail of PK-THPP moves and interacts with I118 (chain B). Simultaneously, the distance between the residues I118 and L239 increases. The interaction between the butanone moiety of PK-THPP and I118 is maintained during the MDs, thus preventing the interaction between I118 and L239, leaving the left fenestration open (Appendix A).

We analyzed how frequently PK-THPP interacts with the hits as well as threonines of the selectivity filter (T93). The contact frequency was calculated by looking at the residues within less than 3 Å distance to PK-THPP. We found that the drug interacts during the entire MDs with residues L122 and L239 in both chains (Figure 6B), and with other residues with less frequency, or not in both chains such as T93, Q126, G236, L244, L247 and T248. Apart from T93, the other residues were identified by alanine scanning as highly significant for the binding (Figure 2C). T93 residue is shown in contact frequency analysis (Figure 6B) because we postulate that it potentiates PK-THPP binding in TASK-3 with respect to the other compounds of the THPP series that cannot establish a hydrogen bond interaction with threonines of the selectivity filter (Figure 6C). Although most of the interactions with the hits are lipophilic (Figure 6C), we also found some other interactions, such as water bridges, with the Q126 residue during the simulation. The butanone moiety was the most mobile part of the drug through the dynamic trajectory (Figure 6D). This moiety is facing the central cavity fenestration interface, interacting from time to time with T93 through hydrogen bonding. Nonetheless, we suggest that this interaction potentiates the binding of compounds of THPP series in TASK-3, and the movement of this group in PK-THPP makes this interaction relevant but not critical for the binding. As a consequence, the T93A mutant does not reduce significantly the sensitivity to PK-THPP in TASK-3 (Figure 2C).

## 3. Discussion

TASK-3 channels were identified as members of the K_2P_ channel family and characterized by Kim et al. [30]. Drug design targeting TASK-3 channels has increased over the years due to their wide expression in central nervous system, their role in several pathological conditions [31], and their overexpression in different cancer types such as breast, gastrointestinal tract, lung and melanoma [32]. Moreover, TASK molecular pharmacology is poorly understood and one of the roadblocks to advancing this understanding lies in the lack of TASK 3D-structures. However, recent K_2P_ channels crystallographic structures (TWIK-1, TRAAK and TREK channels) have allowed the study of the structure of TASK channels through comparative models, which has produced insights on the molecular mechanisms by which these channels are modulated by drugs (Figure 1).

An approach coupling alanine mutagenesis scanning and a computational modeling systematic pipeline has been used to test the hypothesis that PK-THPP blocks TASK-3 similarly to how A1899 blocks TASK-1 [13,18], because both drugs share similar physicochemical characteristics. Besides, the pore of both channels is composed of the same residues, with the exception of residue 247, which is a methionine in TASK-1 and a leucine in TASK-3. To explore the PK-THPP binding site in TASK-3, an alanine scanning mutagenesis was performed. Of the 44 functional TASK-3 pore alanine mutants studied, L122A, Q126A, G231A, G236A, A237V, L239A, L244A, L247A and L248A displayed a significant reduction of PK-THPP inhibitory potency (Figure 2C). Some of these residues are located exclusively in the pore of the channel during the MDs or buried into the protein. However, residues L122, G236 and L239 are at the fenestration-pore interface (Appendix A) and L239 appears to be the key residue that connects both cavities, allowing drug access from the pore to the side-fenestration. Its homologous residue L276 in TRAAK has also been described as a key residue for structural and functional changes between resting and activated TRAAK states, opening a side door to the membrane [33]. In TREK2, L320 (homologous to L239 in TASK-3) was described as part of the norfluoxetine binding site. By using X-ray diffraction, it was demonstrated that in the up state the fenestration is closed by the upward movement and rotation of M4. In the down state, the side chain of L320 is in the side-fenestration providing, with other residues such as F316, a hydrophobic environment close to the selectivity filter in which drugs like Br fluoxetine and norfluoxetine binds [10]. It has also been shown that other drugs, such as A1899, bind to TASK-1 channels by interacting with the pore-lining residues while anchoring in the fenestration [13]. Our results are consistent with those reported for other K_2P_ channels, where the L239 residue and its homologues play a fundamental role in both the up-to-down state transition and the interaction with drugs in the pore and side-fenestration. In TREK-1, for example, L304 (homologous to L239 in TASK-3) interacts with the activator BL-1249 [15].

The PK-THPP binding site was previously explored by Chokshi et al. in TASK-3, who identified L122, L239 and G236 as key residues. Aspartate scanning mutagenesis also suggested that residue V242 is part of the drug binding site [20]. However, TASK3-V242A mutant did not show a significant blockade reduction by PK-THPP in our study. The TASK3-V242D mutant [20] could, for instance, change the behavior of water within the inner pore. In TWIK-1, mutations L146D and L146N, homologous L122 residue of TASK3, revealed that these mutants favored the hydration of the inner pore [12]. TASK3-V242D might lead to an apparent decrease in PK-THPP affinity due to a de-wetting effect at the central cavity more than to a modification of the binding site itself. Aspartate mutants of other hits identified by our alanine scanning mutagenesis such as Q126, A237, L244 and L247 exhibited a greater than 10-fold shift in their PK-THPP IC_50_ relative to wild-type TASK-3; however, they were not defined by Chokshi et al. as part of PK-THPP binding site [20].

To structurally characterize the channel pore and side-fenestration cavities, we modeled TASK-3 with different fenestration states using as templates different K_2P_ channels that were crystalized with the fenestrations closed (TREK-1) and open (TWIK-1 and TREK-2) for the up and down states of the channels, respectively. The fenestration open–closed states of TASK-3 differ not only in the lateral cavities but also in the diameter of the pore (Figure 3A,B), which directly affects how the drug interacts in this cavity. These findings underline the need to study the majority of conformational states of K_2P_ channels when performing computational analyses to predict the binding site and mode of interaction of a given drug since druggable cavities change between states.

After the identification of the residues responsible for the PK-THPP binding, and using the TASK-3 models, we proceeded along a pipeline including a structural-activity relationship (SAR) of THPP series against TASK-3 channel (Figure 4A). SAR allowed us to identify key interaction characteristics not only for PK-THPP, but for other compounds of the THPP series [16]. This revealed that the hydrophobic biphenyl moiety is interacting at the bottom of the central cavity with L244 and L247 residues; the carbonyl group interacts through hydrogen bonds with Q126; the 5,6,7,8 tetrahydropyrido[4,3-d]pyrimidine group interacts at the interface of the side fenestration’s central cavity with L239; and the substituted piperidine in the upper side of the fenestration’s inner cavity (Appendix A, at least four compounds from the five analyzed show these ligand–residue interactions). The last interaction was not identified for compound 17b, which exhibited an IC_50_ 1000-fold lower than PK-THPP because this compound has an unsubstituted pyrrolidine—a small moiety that makes it impossible for 17b at the same time to be anchored to the bottom of the central cavity by hydrophobic interactions through the biphenyl moiety, and to interact with the residues from the upper side of the fenestration–central cavity interface. Besides, a hydrogen acceptor group is not present at the unsubstituted pyrrolidine group in the 17b compound and in consequence it cannot accept hydrogen bonds from threonines of the selectivity filter (e.g., T93 and T199), which is an essential interaction of the potent blockers of TASK channels such as A1899 [13,18] (Figure 4B and Appendix A).

When Coburn et al. [16], synthesized the compounds of THPP series, they realized that lipophilic groups para to the carbonyl was preferred in TASK-3 interactions. However, with large lipophilic moieties, such as the biphenyl, they obtained the best activities against TASK-3, but not with small alkyl groups because they could not cover hydrophobic interactions at the end of the central cavity (L244, L247). The incorporation of a carbonyl linker that could establish a hydrogen bond with Q126 afforded the most promising results [16]. Analogues with a CO linker between the biphenyl and the 5,6,7,8 tetrahydropyrido[4,3-d]pyrimidine group showed good activity in TASK-3 (Appendix A). Diverse substitutions on the 5,6,7,8 tetrahydropyrido[4,3-d]pyrimidine group were not tolerated. Substitution on the saturated ring at position 2 with methyl and phenyl groups resulted in a loss in potency because the 5,6,7,8 tetrahydropyrido[4,3-d]pyrimidine group cannot interact properly with L239, in the reduced interface between the fenestrations and the central cavity. Finally, Coburn et al. [16] found that analogues that possess H-bond acceptor moieties at the 4 position on the saturated ring of the 5,6,7,8 tetrahydropyrido[4,3-d] pyrimidine group were more potent than the unsubstituted ones, due to the H-bonds that this *A* group can establish with the selectivity filter.

The binding mode proposed by SAR studies for compounds of the THPP series, mainly governed by hydrophobic relative ΔG_Bind_ (Table 1 and Table 2), is different than the one proposed by Chokshi et al. [20] for PK-THPP (Appendix A). They propose that the biphenyl group extends pointing toward the selectivity filter, or the biphenyl folds and crosses the pore. However, in our SAR studies, the biphenyl moiety of compounds of THPP series points toward the bottom of the inner cavity. This conformation is also observed in PK-THPP using a wide conformational sampling by massive docking simulations in the 54 poses of PK-THPP obtained in T3twiOO model and 59 poses of PK-THPP obtained in T3tre2OO model within significant clusters. However, in the closed state model of TASK-3 channel (T3tre1CC), the significant cluster poses adopt a similar conformation to the one proposed by Chokshi et al. [20], despite the fact that these authors modeled PK-THPP in an open-state structure of TASK-3 channel based on TWIK-1 (Figure 5).

The binding mode of PK-THPP in Chokshi et al. [20] could even be in line with the mutagenesis study (Figure 2C), where mutation of T248 leads to a decrease of inhibition. With the biphenyl moiety next to the selective filter and the piperidine moiety toward the M4 C-term, PK-THPP could interact with T248 through a hydrogen bond. In only four poses of the massive docking simulations was the piperidin-4-yl-butan-1-one oxygen atom of PK-THPP in close proximity of T248. Despite the proximity of the four poses to T248, none of them interact through a hydrogen bond with this residue (results not shown). In a closed examination of the templates used to model TASK-3, we found that the TASK-3 homologous residues to T248 in TREK-1 (V298), TREK-2 (V324), and TWIK-1 (K278) are not facing the central cavity, which makes it very unlikely that T248 in TASK-3 homology models interacts with PK-THPP through a hydrogen bond.

Based on our massive docking simulations, and in the absence of a determined three-dimensional structure of TASK-3 channel, we consider that conformations exhibiting (*i*) the interaction of the biphenyl moiety at the bottom of the central cavity with L244 and L247, (*ii*) the carbonyl group does so at the middle of the central cavity with Q126, (*iii*) concurrently, the 5,6,7,8 tetrahydropyrido[4,3-d]pyrimidine group does so at the interface of the side-fenestration and the central cavity with L239, while (*iv*) the substituted piperidine is oriented in a way that may accept H-bonds from threonines of the selectivity filter, are required for PK-THPP activity. A similar orientation is adopted by all the compounds of THPP series and the proposed conformation explains the substitution effects observed by Coburn et al. [16]. This binding mode was validated not only by massive docking simulations + MM/GBSA calculations, where the relative ΔG_Bind_ of the representative PK-THPP poses of the significant clusters in T3twiOO and T3tre2OO models was also governed by hydrophobic contribution (Table 4), but also by long unrestricted molecular dynamics simulations where contacts of PK-THPP with residues L239, L244 and L247 (Figure 6B) were present during the entire MDs. Besides these hydrophobic contacts, H-bond interactions with Q126 mediated by water bridges were observed. Water molecules that establish the water bridges are placed on the central cavity. Also, a H-bond is present between PK-THPP and T93 (Figure 6C), although this part of the molecule is very mobile (Figure 6D) and the H-bond interaction is not permanent during the MDs. Specifically, atom 32 (oxygen) of PK-THPP establishes this interaction and atom 35 (carbon), which is even more mobile than oxygen 35, establishes a hydrophobic interaction with I118 (chain B). Simultaneously, the distance between residues I118 (chain B) and L239 (chain A) increases, opening a side door to the membrane (Figure 6D, Appendix A), suggesting a crucial role of L239 in structural and functional changes between resting and activated TASK-1 states. Residue L239 at the fenestration–pore interface interacts in TASK channels with pore blockers such as A1899 [18] and PK-THPP [20], but also with allosteric drugs such as bupivacaine, which blocks by binding to the side fenestrations exclusively [14].

The TASK-3 inhibitor PK-THPP studied here is a potent molecular tool to gain insights regarding the TASK-3 channels pharmacology. PK-THPP blocks the closest relative of TASK-3 (TASK-1) with almost nine-fold less potency [14]. They shared the binding site, except residue L247 which is M247 in TASK-1. Differences in the block of TASK channels by PK-THPP could be regulated by residue 247 but also the C-terminal domain of these channels, which share little homology, controlling the pore access to PK-THPP.

The theoretical approach applied confirmed the hits identified by our alanine mutagenesis scan and suggest how small structural differences between THPP compounds results in a wide biological activity spectrum. Since K_2P_ channels have arisen as novel drug targets, a detailed understanding of how inhibitors interact with TASK-3 will facilitate optimization for basic research in physiology and pharmacology. Commanding future studies on the molecular pharmacology of other K_2P_ channels, our results might help to unravel the role of the side-fenestrations in the modulation of these channels, improving rational design to increase the drug potency and selectivity against K_2P_ channels.

## 4. Materials and Methods

### 4.1. Oocyte Preparation, cRNA Synthesis, and Injection

Oocytes from *Xenopus laevis* were prepared and injected with the synthetized cRNA as previously described [16]. Briefly, oocytes were incubated in OR2 solution (NaCl 82.5 mM, KCl 2 mM, MgCl_2_ 1 mM, and HEPES 5 mM, (pH 7.5), supplemented with 2 mg/mL collagenase II) and stored at 18 °C in ND96 solution (NaCl 96 mM, KCl 2 mM, CaCl_2_ 1.8 mM, MgCl_2_ 1 mM, and HEPES 5 mM (pH 7.5), supplemented with 33.6 μM gentamycine, 2.5 mM sodium pyruvate, and 0.5 mM theophylline). The pSGEM expression vector was used to clone human TASK-3 (*KCNK9*, NM_001282534.2). The QuikChange™ Site Directed Mutagenesis Kit (Stratagene, La Jolla, CA, USA) was used to introduce site direct mutations. Subsequently, cDNA was linearized and cRNA was synthesized with the mMESSAGE mMACHINE-Kit (Ambion, Foster City, CA, USA). Finally, oocytes were each injected with 50 nL (5 ng) of cRNA.

### 4.2. Electrophysiology

All two-electrode voltage clamp (TEVC) measurements were performed as previously described [18]. Briefly, recordings were performed at room temperature 48 h after cRNA injection with a Digidata 1200 series (Axon Instruments, Union City, CA, USA) as analog to digital (A/D) converter and a TurboTEC 10CD (npi) amplifier. Borosilicate glass capillaries GB 150TF-8P (Science Products, Hofheim, Germany) were pulled with a DMZ-Universal Puller (Zeitz, Martinsried, Germany). Recording glass capillaries had a resistance of 0.5−1.5 MΩ and were filled with 3 M KCl solution, whereas ND96 was used as extracellular recording solution. With the following protocol, inhibition by 400 nM THPP was analyzed: a test pulse to 0 mV of 1 s duration from a holding potential of −80 mV, followed by a voltage step to −80 mV for 1 s, and directly followed by another 1 s test pulse to +40 mV. The sweep time interval was 10 s. For IC50 measurements, the following PK-THPP concentrations were used in the recording solution: 50 nM, 200 nM, 500 nM, 1000 nM and 10,000 nM. Three to seven oocytes were used for each experiment.

### 4.3. TASK-3 Modeling

Since the three-dimensional (3D) structure of TASK-3 has not been solved, the sequence of human TASK-3 was downloaded (UniProtKB accession number: Q9NPC2) and three homology models were built using the following crystal structures as templates: TREK-2 (PDB: 4BW5), TREK-1 (PDB: 4TWK) and TWIK-1 (PDB: 3UKM). These structures have differences in the fenestration states (they could be open or closed); therefore, the different TASK-3 models were used to study the interactions between compounds of THPP series and TASK-3 with diverse fenestrations characteristics. The TASK-3 homology models were built according to the multiple sequence alignment published by Brohawn et al. [5] and optimized using Prime software [34,35]. The models (Appendix A) were named according to the template and the fenestration state as follows: T3tre2OO (TASK-3 built from TREK-2 in Open-Open fenestration state), T3tre1CC (TASK-3 built from TREK-1 in Close-Close fenestration state), and T3twiOO (TASK-3 built from TWIK-1 in Open-Open fenestration state). Models were built as monomers and assembled as dimers using Maestro software [36]. The quality and internal consistence of 3D models were evaluated by using the PROCHECK package [37]. Two K^+^ ions were associated to the models in positions S_2_ and S_4_ of the selectivity filter and two water molecules at sites S_1_ and S_3_ (Figure 1).

To prepare the systems, Maestro version 9.2 software was used for adding hydrogen atoms, to assign the bonds order and partial charges to the homology models. Following this, they were embedded into a pre-equilibrated phosphatidyl oleoyl phosphatidylcholine (POPC) bilayer in a periodic boundary condition box with pre-equilibrated SPC water molecules. Finally, the systems were neutralized by adding K^+^ counter ions to balance the net charge of the systems and KCl at a concentration of 0.096 M was added to simulate physiological conditions of the channel.

Each system was subjected to a conjugate gradient energy minimization, then the atomistic systems were equilibrated with a spring constant force of 1.0 kcal × mol^−1^ × Å^−2^ applied to the secondary structure for 25 ns at constant pressure (1 atm), temperature (300 K), and number of atoms using the isothermal-isobaric ensemble and the Nosé–Hoover method with a relaxation time of 1 ps applying the MTK algorithm [38], employing the OPLS-AA force field in Desmond v3.0 program [36,39] with a timestep of 4 fs. The TASK-3 model stability during the MDs was validated by calculating the RMSD. To analyze the intracellular pore as well as the fenestrations in the models the algorithm HOLE was used [40].

### 4.4. Modeling of Compounds of THPP Series

The derivatives based on a 5,6,7,8-tetrahydropyrido[4,3-d]pyrimidine [16] with a reported IC_50_ < 30 µM (Appendix A) were sketched using Maestro suite. The compound 23 is named as PK-THPP in this investigation. The compounds were optimized and processed using LigPrep [36] with the force field OPLS_2005 [41] to obtain the equilibrium geometry, the geometrical parameters and the potential energies surfaces.

### 4.5. Molecular Docking

A molecular docking was performed in order to study how the 29 compounds of the THPP series interact with TASK-3 models (Appendix A). The structure of TASK-3 models was taken from the last frame of the 25 ns MDs. The center of the grid box was focused into the residues L122 and L239 because their alanine mutants reduce the percentage of inhibition at least twice for PK-THPP [20], the most potent TASK-3 channel antagonist of the THPP series. Molecular docking runs were performed using a grid box of (30 × 30 × 30) Å centered into the residues L122 and L239, with the standard precision (SP) scoring function in the Glide v5.7 software [36,42]. The top-10 poses per docking simulation were used for further analysis.

To find the most probably PK-THPP binding mode in TASK-3 models, and taking into account that the incorporation of conformational rearrangements in the channel binding pocket into predictions of the ligand binding pose is critical to improve docking results [43,44], we carried out several dockings of PK-THPP into the structures collected from the last 10 ns of the 25 ns MDs of TASK-3 homology models using the parameters described above, obtaining the top 10 poses per docking simulation. At the end, we obtained 100 poses (10 poses for each frame, 10 frames for each model) per model.

### 4.6. Relative Binding Free Energy calculations (Relative $\Delta G_{bind}$)

The computational method Molecular Mechanics-Generalized Born Surface Area (MM-GBSA), which combines molecular mechanics energy and implicit solvation models [45], was employed using Prime [36] after the docking of the 29 compounds of THPP series into TASK-3 models to rescore and correlate their reported IC_50_ and the predicted $\Delta G_{bind}$ against the channel. The aim was to identify the TASK-3 homology model that presents the best experimental-theoretical correlation. In MM-GBSA, the binding free energy between the ligand and the receptor (TASK-3 channel) to form a complex is calculated as
(1)$$\Delta G_{bind}=\Delta H-T\Delta S\approx \Delta E_{MM}+\Delta G_{sol}-T\Delta S$$
(2)$$\Delta E_{MM}=\Delta E_{internal}+\Delta E_{electrostatic}+\Delta E_{vdw}$$
(3)$$\Delta G_{sol}=\Delta G_{PB/GB}+\Delta G_{SA}$$
where ΔE_MM_ corresponds to the molecular mechanics energy change and includes internal (bond, angle and dihedral energies), electrostatic, and van der Waals energies changes; ΔG_sol_ is the solvation free energy change, which corresponds to the sum of electrostatic solvation energy ΔG_PB/GB_ (polar contribution) and non-electrostatic solvation component ΔG_SA_ (non-polar contribution). The polar contribution was calculated using the Generalized Born model, while the non-polar energy was calculated by solvent accessible surface area (SASA) [46,47]. Finally, ΔS is the change of conformational entropy upon binding at a temperature *T*. Corrections for entropic changes were not applied because we used compounds derived from the same scaffold (5,6,7,8 tetrahydropyrido[4,3-d]pyrimidine) in all calculations. Besides, it has been previously reported that the lack of an entropy evaluation is not critical for calculating the MM-GBSA free energies for similar systems [48,49,50,51]. The VSGB solvation model [52] and OPLS-2005 force field were employed to accomplish the calculations. Residues located at 5 Å from the ligands were included in the flexible region, and all other TASK-3 atoms were kept frozen. The computed relative binding free energies were correlated against experimental IC_50_ values for the 29 compounds of the THPP series selected in this study. The degree of statistical correlation (*R*^2^) between the both experimental IC_50_ and computed relative ΔG_Bind_ values is reported.

### 4.7. Molecular Pharmacophore Modeling

Phase from Schrödinger molecular modeling Suite [36] was used to create the pharmacophore. Phase classifies the chemical features of the ligands as hydrogen bond acceptors (*A*), aromatic rings (*R*), hydrophobic groups (*H*), hydrogen bond donors (*D*), negatively charged groups (*N*) and positively charged groups (*P*). In order to develop a shared pharmacophore, Phase evaluates the n-point pharmacophores resulting from the conformational sets of active compounds and then finds all 3D arrangements with pharmacophore qualities common for these compounds [53].

### 4.8. Clustering of Conformers of PK-THPP

For each TASK-3 model 100 poses were obtained (10 poses for each frame, 10 frame for each model). To process and to organize the 300 poses we used the Conformer Cluster script (available at www.schrodinger.com/scripcenter/) as described previously [13]. The script builds an RMSD distance matrix [54] between pairs of corresponding atoms, followed by an optimal rigid-body superposition [55]. Atomic RMSD values were calculated between heavy atoms from PK-THPP, and the linkage average method was used to cluster the PK-THPP poses.

### 4.9. Molecular Dynamics Simulations (MDs)

The PK-THPP–T3tre2OO complex, which presents the best correlation between the reported IC_50_ and the predicted relative ΔG_Bind,_ was subjected to a conjugate gradient energy minimization and 275 ns MDs in Desmond v3.0 using OPLS-2005 force field [56,57]. The system was built as described above (see Section 4.3). The first 25 ns were performed applying a restraint spring constant of 0.5 kcal ^×^ mol^−1 ×^ Å^−2^ to the secondary structure of the channel, then the last frame was taken and a second non-restricted 250 ns MDs was performed. For the TASK-3 homology model T3tre2OO without ligand, a 250 ns MDs was performed following the same protocol previously described.

### 4.10. Reagents

PK-THPP was custom synthesized by Aberjona Laboratories (Beverly, MA, USA). The compound was solubilized in dimethylsulfoxide (DMSO) as a 10 mM stock. It has been reported that TASK-3 function is not affected by DMSO at bath concentrations up to 1% [20] and our electrophysiological measurements did not exceed this level.

## Figures and Tables

**Figure 1 ijms-20-02252-f001:**
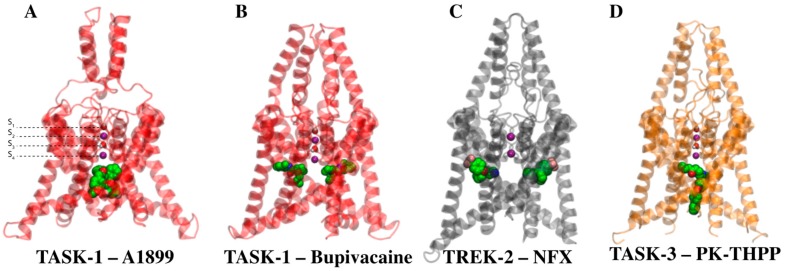
Binding site of different drugs in K_2P_ channels. (**A**) A1899 blocker interacting with TASK-1 at the central cavity [13]. (**B**) Local anesthetic bupivacaine allosterically inhibiting TASK-1 channels interacting in the lateral fenestration [14]. (**C**) Norfluoxetine interacting with TREK-2 in the lateral fenestrations [10]. (**D**) PK-THPP blocker interacting in the central cavity with TASK-3 (reported in this study). Two K^+^ ions were associated to the TASK homology models in positions S_2_ and S_4_ of the selectivity filter and two water molecules at sites S_1_ and S_3_ [13,14, current study].

**Figure 2 ijms-20-02252-f002:**
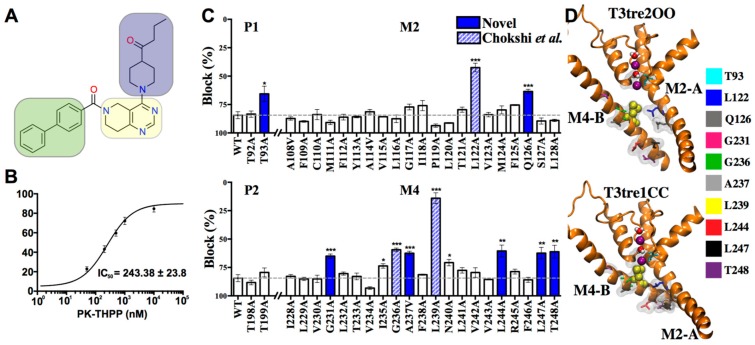
PK-THPP biding site. (**A**) Chemical structure of PK-THPP, 5,6,7,8 tetrahydropyrido[4,3-d]pyrimidine (yellow), piperidin-4-yl-butan-1-one (blue), biphenyl (green). Yellow and green moieties are separated by a carbonyl group (**B**) Dose-response curve of PK-THPP on human TASK-3. Block was analyzed at the end of the test pulse of +40 mV. (**C**) TASK-3 mutant channels with a reduced sensitivity to PK-THPP are marked with asterisks. All values are expressed as means ± standard error (S.E., bars). The significance of the differences between the mutants and the WT result was assessed using two-tailed Student’s t-tests. Asterisks indicate significance: * *p* < 0.05; ** *p* < 0.01; *** *p* < 0.001. Other differences did not reach statistical significance. For all oocyte experiments, *n* = 3–7 cells. Residues identified as THPP binding site hits have significant levels of *p* < 0.01 and *p* < 0.001, except T93 (*p* < 0.05); this is important from the modelling point of view. (**D**) Residues of PK-THPP binding site are facing the pore and the fenestrations as it is shown in TASK-3 models in the down (T3tre2OO) and up (T3tre1CC) states (lateral view from the pore). Hits (Chain A: T93, L122 and Q126; Chain B: G231, G236, A237, L247 and T248) are represented in licorice with individual colors. L239 hit, which could play a fundamental role in the up-to-down state transition is represented using yellow Van der Waals spheres. For better visualization, only transmembrane (TM) segments M2-chain A and M4-chain B are represented.

**Figure 3 ijms-20-02252-f003:**
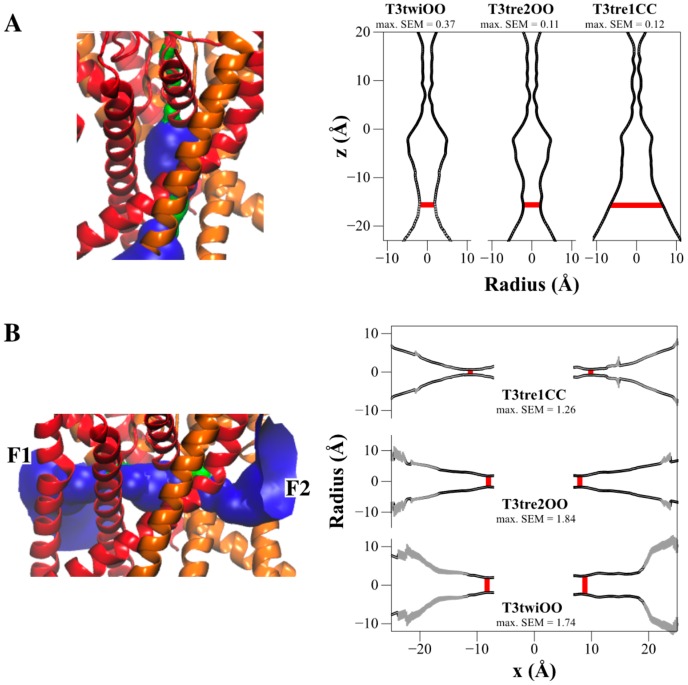
TASK-3 modeling. (**A**) Left: HOLE representation of the pore for the T3tre2OO model. Right: Graphs show the average diameter and the standard error of mean (SEM) of the pore in T3tre1CC, T3tre2OO and T3twiOO models for the last 10ns MDs (10 structures from each MDs → 1 frame per ns were taken). The central cavity is located at the bottom of the selectivity filter (−20 Å to 0 Å in the *z*-axis). The z-axis is lined by the ion-conducting pore. Within the central cavity, the bottleneck of models with open fenestrations is at z-axis = −15.5 Å (size 4.32 ± 0.013) in T3tre2OO and at *z*-axis = −15 Å (size 3.76 ± 0.263) in T3twiOO. Meanwhile, at *z*-axis = −15.5 Å, the pore size in T3tre1CC is 12.96 ± 0.061, red line. (**B**) Left: HOLE representation of the fenestrations (F1 left fenestration, F2 right fenestration) for the T3tre2OO model. Right: Plot of the radius of the TASK-3 fenestrations for each MD. The bottleneck diameters (red line) of F1 and F2 are at the following positions in the *x*-axis in each model: T3tre1CC, F1 x-axis = −11 Å (size 1.34 ± 0.178) and F2 x-axis = 10 Å (size 1.50 ± 0.153); T3tre2OO, F1 x-axis = −8 Å (size 3.54 ± 0.093) and F2 x-axis = 7.5 Å (size 3.6 ± 0.035); T3twiOO: F1 *x*-axis = −8 Å (size 3.84 ± 0.096) and F2 x-axis = 8 Å (size 4.66 ± 0.052). The x-axis is perpendicular to the ion-conducting pore and is lined by the access to the hydrophobic core of the lipid bilayer from the pore central cavity. HOLE color code used is as follows: Blue, radius > 1.15 Å; green, radius between 0.6–1.15 Å. Subunits A (orange) and B (red) of the homology model of TASK-3 channel based on TREK-2 (T3tre2OO) are shown in cartoon representation in each left figure.

**Figure 4 ijms-20-02252-f004:**
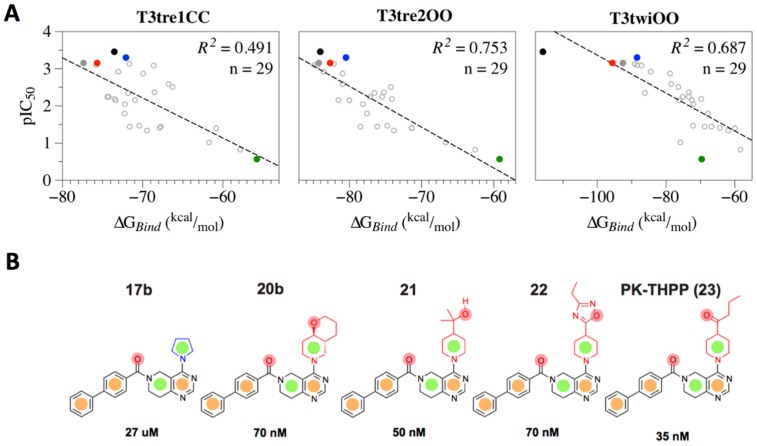
TASK-3–THPP series structural-activity correlation. (**A**) Correlation plots obtained for THPP series interacting with TASK-3 models in down-state (T3tre1CC) and up-state (T3twiOO and T3tre2OO). The compound docking poses with the best relation between the relative binding free energy (ΔG_Bind_) per model and their experimental biological activity (pIC_50_), expressed as Ln (100/IC_50_), were used for comparison. Five compounds of the THPP series (dots) were selected to understand how slight structural modifications result in significant differences in the binding affinity against TASK-3. They are PK-THPP (black), 17b (green), 20b (gray), 21 (blue) and 22 (red). (**B**) Chemical structures of the five studied ligands of THPP series. The low affinity compound 17b exhibits a different moiety (blue, unsubstituted pyrrolidine) than the high affinity compounds from 20b to 23 (red, substituted piperidine). Consequently, the shared seven-point pharmacophore *RRAHRHA* (from the left to the right and upper side of the molecules) where *R* (aromatic ring, orange dot), *A* (H-Bond acceptor group, red dot) and *H* (hydrophobic group, green dot) shared by the high affinity compounds is not present completely in the low affinity compound 17b, which does not exhibit the last *A* group.

**Figure 5 ijms-20-02252-f005:**
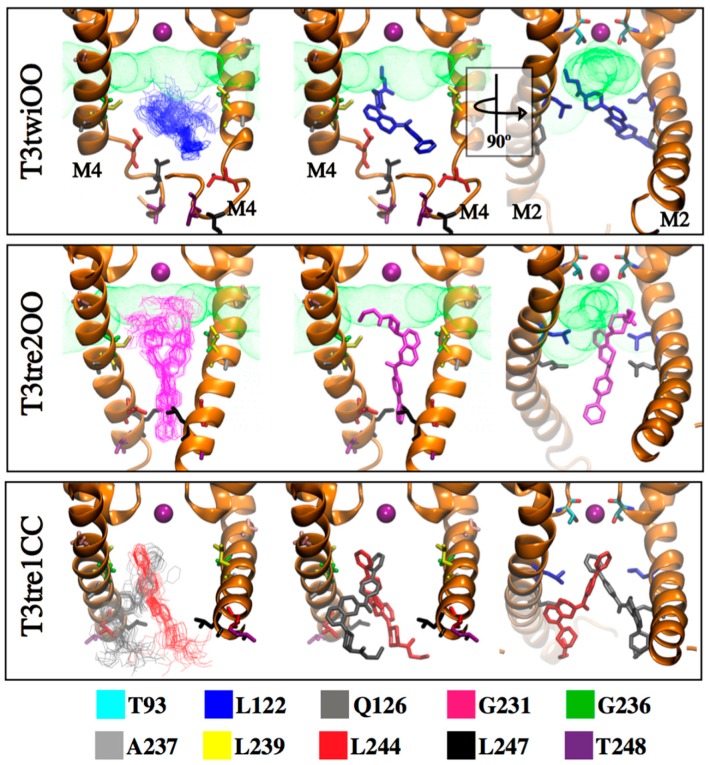
Orientation of the PK-THPP poses within the significant clusters. Significant cluster-10 (blue), −23 (magenta), −28 (gray) and −33 (red) are represented in sticks. K^+^ ions at S_4_ are shown in sphere representation and TASK-3 models in cartoon representation. Right, the structure nearest the centroid in each significant cluster per model is shown. Hits within 5 Å of the poses are shown in stick representation as well as T93 residue. For better visualization, TM3 and TM4 are not shown. Open fenestration in T3twiOO and T3tre2OO models are shown as green dotted surface.

**Figure 6 ijms-20-02252-f006:**
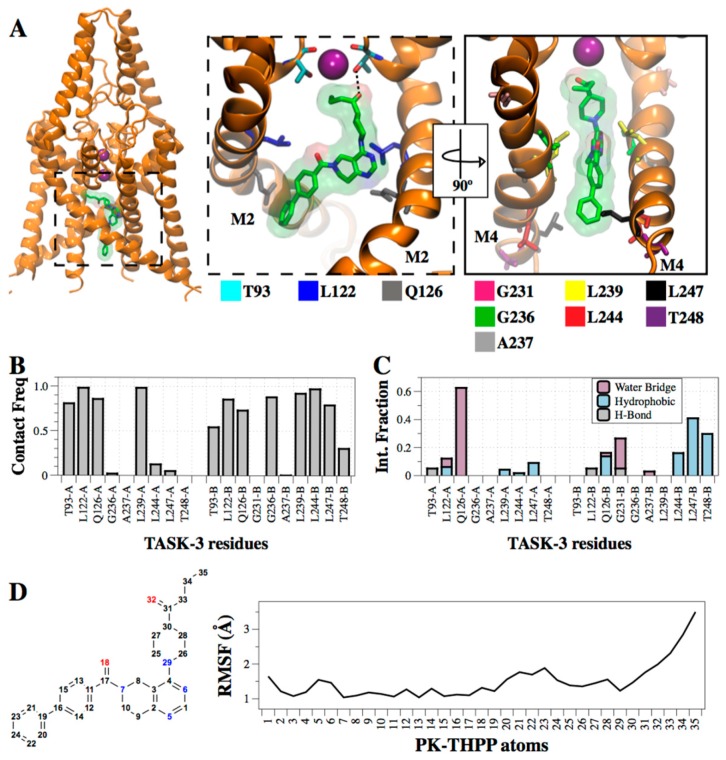
Redefined binding mode of PK-THPP in TASK-3. (**A**) Left, PK-THPP interacting with TASK-3. Right, zoomed binding site with key residues represented as stick and the hydrogen bond with T93 shown explicitly. (**B**) Contact frequencies of PK-THPP with T3tre2OO residues at 3 Å. Bars indicate the contact frequency along the 250ns MDs. (**C**) Interactions between the residues of T3tre2OO and PK-THPP are categorized into hydrophobic, water bridges and hydrogen bonds. The stacked bar charts are normalized over the course of the unrestrained MDs. The interactions were calculating using the Simulation Interaction Diagram tool included in the Schrödinger Suite. (**D**) Root mean square fluctuation (RMSF, black line) characterizing the internal atom fluctuations of PK-THPP during the 250ns MDs. The numbers for the atoms are presented in the *x*-axis and in the structure.

**Table 1 ijms-20-02252-t001:** MM-GBSA terms calculated for THPP series compounds–3tre2OO complexes.

THPP Series	IC_50_ (µM)	ΔE_MM_				ΔG_sol_		ΔG_Bind_
		ΔE_internal_	ΔE_electrostatic_		ΔE_vdw_	ΔG_GB_	ΔG_SA_	
			Coulomb	Hbond				
PK-THPP	0.035	11.37	−6.48	−0.40	−57.95	13.21	-43.79	–84.05
21	0.05	5.88	−22.14	0.67	−49.63	27.08	–42.34	–80.49
20b	0.07	9.27	−6.44	−2.03	−50.21	5.07	–39.93	–84.26
22	0.07	14.81	−20.47	−1.07	−55.36	18.19	–38.79	–82.69
17b	27	6.45	−5.58	−1.51	−39.99	13.60	–32.19	–59.21

MM-GBSA terms (in kcal/mol) are explained in Equations (1)–(3).

**Table 2 ijms-20-02252-t002:** Nature of the interactions of compounds of THPP series with the hits. Atomic interactions can be visualized in detail in Appendix A.

	# of Interactions (# by Chemical Groups)		
Ligand Name	Hydrophobic	Polar	Hydrogen bond
17b	5 (3→ Biphenyl, 1→ Pyrimidine, 1→ Substituent)	1 (Biphenyl)	-
20b	6 (4 → Biphenyl, 1→ Pyrimidine, 1→ Substituent)	1 (Biphenyl)	2 (1→Carbonyl, 1 Pyrimidine)
21	4 (2→ Biphenyl, 1→ Tetrahydropyridine, 1→ Pyrimidine)	-	1 (Carbonyl)
22	2 (1→ Biphenyl, 1→Substituent)	-	2 (1→ Carbonyl, 1 Pyrimidine)
23	5 (3→ Biphenyl, 2→ Pyrimidine)	1 (Substituent)	1 (Carbonyl)

**Table 3 ijms-20-02252-t003:** Best Clusters of PK-THPP poses.

T3twiOO		T3tre2OO		T3tre1CC	
No.	Pop.	No.	Pop.	No.	Pop.
10	54	23	59	28	25
				33	26

**No.**, number of clusters; **Pop.**, population.

**Table 4 ijms-20-02252-t004:** MM-GBSA terms calculated for PK-THPP–TASK-3 complexes

PK-THPP Pose	TASK-3 Model	ΔE_MM_				ΔG_sol_		Relative ΔG_Bind_
		ΔE_internal_	ΔE_electrostatic_		ΔE_vdw_	ΔG_GB_	ΔG_SA_	
			Coulomb	Hbond				
84	T3twiOO	25.2	−20.0	−1.1	−63.7	18.6	−75.1	−115.8
177	T3tre2OO	11.4	−6.5	−0.4	−57.9	13.2	−43.8	−84.1
270	T3tre1CC	6.3	−21.8	−1.1	−51.9	34.6	−39.7	−73.5

MM-GBSA terms (kcal/mol) are explained in Equations (1)–(3).

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
