# Peer review of "Structure/Activity Analysis of TASK-3 Channel Antagonists Based on a 5,6,7,8 tetrahydropyrido[4,3-d]pyrimidine"

_ijms, 2019, doi:10.3390/ijms20092252_

Round 1
Reviewer 1 Report
In their manuscript entitled structure/activity analysis of TASK-3 channel antagonists based on a 5,6,7,8 tetrahydropyrido[4,3-d]pyrimidine the authors combine functional assays with modeling to identify key residues involved in the binding of blockers of TASK3 channel protein.
The topic is of interest and should get attention from the community. The authors have synergized complementary technics to support their claims. However, this paper must be improved before it can be published.
Major comments:
One of my major comments is the claim about the importance of T93 and T199 for the binding of blocker (line 80 ‘is due to the presence of a hydrogen bond acceptor group that can establish interactions with the threonines of the selectivity filters’ and line 227: ‘is an essential interaction of the potent blockers of TASK channels’). The in vitro data clearly do not support this claim. The threonine to alanine mutations induce no change of the IC50 values (figure 1C). Moreover, the results from the MD simulations show that the ligand is having almost no specific interaction (Hbond) with residue T93 (Figure 6E). Then the conclusion ‘we suggest that this interaction is essential to potentiate the binding of compounds of THPP’(line 332) must be removed from the paper.
On the other side of the pore, mutation of threonine 248 leads to a decrease of inhibition. This result strengthens the hypothesis of a binding pose with the biphenyl next to the selective filters and the piperidin moiety interacting with T248 through a hydrogen bond. This docking pose seems to be more in line with the in vitro data.
The introduction is not well written. The authors should add a figure (similar to the one they provide in REF13, or in figure 2.A) to summarize the structural knowledge of TASK-3. I think that the paragraph (l 384 to 398) should be added in the introduction. This part is important to understand the difference from the REF23 study. A special effort should be done to clearly distinguish published results from new one.
The figures, while numerous, are not clear enough. The authors should simplify the figures and focus on the most important points. The authors often refer to their previous study REF13. They should add a structure of the complex TASK1-A1899 in the introduction to summarize the knowledge of the binding of K2p blockers.
The structures of the studied ligands are presented in Figure 4A. They should appear earlier in the paper. Moreover, the two K+ ions are only mentioned in the caption of Figure 5 while they already appear in Figure 1D. The authors only mentioned these two ions and two water molecules added to specific position S1, S2, S3 and S4 line 528,529. Are these water molecules involved in the binding as water bridge? The authors should add S positions on the full structure of TASK3.
The authors should emphasize the results from the in vitro data. The location of the mutated amino acids should be improved (Figure 1D). The authors should also comment on the gain of function. Does the mutation P119A increased the receptor inhibition compared to WT?
Technical limitation
The rmsd from the 25 first ns is calculated on a constraint structure (1 kcal.mol.A-2 , line 542). Thus, it is obvious that the structure should remain close to its the initial conformation. The authors have taken 10 structures from this MD to consider the flexibility of the receptors. A constraint 25ns MD simulation cannot achieve a ’wide conformational sampling’ (line 444). Why not using a flexible docking procedure instead?
The authors have made three models of TASK3. They should take more care for describing the difference between the three structures. The Hole plots are not clear. The authors must specify what are the meaning of z and x axis. Moreover, they should add numbers to quantify the difference in pore size between the three models.
The authors have decided to correlate IC50 to their theoretical binding affinity. While the correlation is good and allow to choose among the different models of TASK3, the EQ4 appears to be false. The authors should be careful when comparing binding affinity to IC50 (line220). IC50 and affinity are not always equal (see the paper from Yung-ChiChengWilliam H.Prusoff: Relationship between the inhibition constant (KI) and the concentration of inhibitor which causes 50 per cent inhibition (I50) of an enzymatic reaction)
Minor point.
Table 1 does not bring any input for the paper. It should be provided as supplementary material.
The author should use the term ΔMM-GBSA instead of ΔGbind while they are not calculating any entropy penalty.
Line 133 ’25 ns MDs to stabilize them’. MD simulation do not stabilize a molecule. A MD simulation samples the conformational space of the system.
The authors should be more precise on the protein RMSD calculation by providing the atoms they were choosing to calculate it.
Left panel from figure 5 is useless for the paper. Instead of a color code, the authors should label the amino acids on the right panel of figure 5.
The paragraph between line 425 and 438 seems to be out of the scope of this paper as it is a comment on a previous paper (REF14).
The pharmacophore study is bringing no new insight for this study as the ‘compounds derived from the same scaffold’ (line 587 588). The authors are solely focusing on the lack of H-bond acceptor on the compound 17 which can be directly observed on the molecular structure.
Line 624: 300°K should be 300 K.
Author Response
Response to Reviewer 1 Comments
In their manuscript entitled structure/activity analysis of TASK-3 channel antagonists based on a 5,6,7,8 tetrahydropyrido[4,3-d]pyrimidine the authors combine functional assays with modeling to identify key residues involved in the binding of blockers of TASK3 channel protein.
The topic is of interest and should get attention from the community. The authors have synergized complementary technics to support their claims. However, this paper must be improved before it can be published.
Major comments:
One of my major comments is the claim about the importance of T93 and T199 for the binding of blocker (line 80 ‘is due to the presence of a hydrogen bond acceptor group that can establish interactions with the threonines of the selectivity filters’ and line 227: ‘is an essential interaction of the potent blockers of TASK channels’). The in vitro data clearly do not support this claim. The threonine to alanine mutations induce no change of the IC50 values (figure 1C). Moreover, the results from the MD simulations show that the ligand is having almost no specific interaction (Hbond) with residue T93 (Figure 6E). Then the conclusion ‘we suggest that this interaction is essential to potentiate the binding of compounds of THPP’(line 332) must be removed from the paper.
à We thank the reviewer for this observation. We changed the statement ‘we suggest that this interaction is essential to potentiate the binding of compounds of THPP’ by `we suggest that this interaction is essential to potentiates the binding of compounds of THPP series in TASK-3´. However, this statement most be read in the framework of the whole appreciation, which extends now from line 358 to line 358: `Nonetheless, we suggest that this interaction potentiates the binding of compounds of THPP series in TASK-3, the movement of this group in PK-THPP makes this interaction relevant but not critical for the binding. In consequence, T93A mutant does not reduce significantly the sensitivity to PK-THPP in TASK-3 (Figure 2C)´.
On the other side of the pore, mutation of threonine 248 leads to a decrease of inhibition. This result strengthens the hypothesis of a binding pose with the biphenyl next to the selective filters and the piperidin moiety interacting with T248 through a hydrogen bond. This docking pose seems to be more in line with the in vitro data.
à In our structural-activity relationship (SAR) studies, the biphenyl moiety of analyzed compounds ofTHPP series points toward the bottom of the inner cavity (Figure S2). This conformers orientation is also observed in PK-THPP using a wide conformational sampling by massive docking simulations in the 54 poses of PK-THPP obtained in T3twiOO model and the 59 poses of PK-THPP obtained in T3tre2OO model within significant clusters. Only in the closed state model of TASK-3 channel (T3tre1CC), the PK-THPP poses within the significant clusters adopt conformations with the biphenyl next to the selective filter (Figure 5).
Based on the referee´s comment, we searched in silico a hydrogen bond interaction between the piperidine of PK-THPP and threonine 248. The search was done using an in house script. From the 100 docking poses of PK-THPP generated in each homology model (T3twiOO, T3tre2OO and T3tre1CC) were selected those with the piperidin-4-yl-butan-1-one oxygen atom in close proximity (5 Å) of any atom of T248 residue.
Although the significant cluster of PK-THPP poses in T3tre1CC adopt a conformation with the biphenyl next to the selective filter, the TM4 C-term is very open in this model and T248 is far from PK-THPP. As reference of pore size in T3tre1CC, check the figure 3B. In consequence, it does not exist a single pose of PK-THPP in T3tre1CC model in close proximity to T248.
Only in four poses of the open state models of TASK-3 channel (T3twiOO and T3tre2OO models), with a narrower pore diameter than T3tre1CC (Figure 3B), the piperidin-4-yl-butan-1-one oxygen atom of PK-THPP is in close proximity of T248. One of these poses is in T3tre2OO model and the rest in T3twiOO model. This is not surprising, because the template used to build the T3twiOO model (TWIK-1, PDB ID: 3UKM) has an interfacial C helix at TM4 parallel to the membrane that closes the pore. Although the proximity of the four poses of PK-THPP to T248, none of them interact through a hydrogen bond with this residue. In the next table, we summarize the distances between the piperidin-4-yl-butan-1-one oxygen atom of PK-THPP and the T248 gamma oxygen in each pose.
Pose* | Distance (Å) |
T3tre2OO_53 | 6.6 |
T3twiOO_64 | 9.3 |
T3twiOO_65 | 6.2 |
T3twiOO_68 | 7.4 |
* The complexes (PKTHPP-TASK3) of these poses are available as pdb files for Reviewer 1.
In addition, in a closed examination of the templates used to model TASK-3 with different fenestration states, we found that the TASK-3 homologous residues to T248 in TREK-1 (V298), TREK-2 (V324), and TWIK-1 (K278) are not facing the central cavity, which makes it very unlikely that T248 in TASK-3 homology models interacts with PK-THPP through a hydrogen bond.
Founded on this analysis we wrote in the manuscript (lines 467 to 483):
“Chokshi et al. [23] binding mode of PK-THPP, even could be in line with the mutagenesis study (Figure 2C), where mutation of T248 leads to a decrease of inhibition. With the biphenyl moiety next to the selective filter and the piperidine moiety toward the M4 C-term, PK-THPP could interact with T248 through a hydrogen bond. Only in four poses of the massive docking simulations, the piperidin-4-yl-butan-1-one oxygen atom of PK-THPP is in close proximity of T248. Although the proximity of the four poses to T248, none of them interact through a hydrogen bond with this residue (results not shown). In a closed examination of the templates used to model TASK-3, we found that the TASK-3 homologous residues to T248 in TREK-1 (V298), TREK-2 (V324), and TWIK-1 (K278) are not facing the central cavity, which makes it very unlikely that T248 in TASK-3 homology models interacts with PK-THPP through a hydrogen bond.
Based in our massive docking simulations and in the absence of a determined three-dimensional structure of TASK-3 channel, we consider that conformations exhibiting (1) the interaction of the biphenyl moiety at the bottom of the central cavity with L244 and L247, (2) the carbonyl group does so at the middle of the central cavity with Q126, (3) at the same time, the 5,6,7,8 tetrahydropyrido[4,3-d]pyrimidine group does so at the interface of the side-fenestration and the central cavity with L239 while (4) the substituted piperidine is oriented in a way that may accept H-bonds from threonines of the selectivity filter, are required for PK-THPP activity.”
The introduction is not well written. The authors should add a figure (similar to the one they provide in REF13, or in figure 2.A) to summarize the structural knowledge of TASK-3. I think that the paragraph (l 384 to 398) should be added in the introduction. This part is important to understand the difference from the REF23 study. A special effort should be done to clearly distinguish published results from new one.
à We thank the reviewer for the comment, however we consider that the structural knowledge of TASK-3 is scarce and a figure could have only the structure as in figure 2A and nothing else. For a better comprehension, we added a figure to the introduction similar to figure 2A but with the blockers of K2P channels where the binding site has been identified. Also, the paragraph (line 384 to 398) was added partially to the introduction. We also made a special effort in the introduction to distinguish published results (Ex: Chokshi et al., 2015 results [REF20]) from our results.
The figures, while numerous, are not clear enough. The authors should simplify the figures and focus on the most important points. The authors often refer to their previous study REF13. They should add a structure of the complex TASK1-A1899 in the introduction to summarize the knowledge of the binding of K2p blockers.
à Thank for the comment. We describe point by point what we did according to your suggestion:
1) We added the complex TASK1-A1899 to the new Figure 1.
2) Figure 3 (previously Figure 2) is now simplified and the time dependence of the RMSD backbone of TASK-3 models during 25ns MDs is in the Supplementary Material.
3) In figure 4 we merged previous figure 3 and figure 4A. Previous figures 4B and 4C are now in the Supplementary Material.
4) Figure 5 contains now only the significant clusters represented is sticks and the structure nearest the centroid in each significant cluster per model. The matrices are now in the Supplementary Material.
5) In figure 6, we removed the time dependence of the RMSD for PK-THPP atoms and TASK-3 backbone atoms during the 250 ns unrestrained MDs. Also, the hole profile before and after 250 ns MD simulations in the presence of PK-THPP was removed. This information is now in the Supplementary Material.
The structures of the studied ligands are presented in Figure 4A. They should appear earlier in the paper. Moreover, the two K+ ions are only mentioned in the caption of Figure 5 while they already appear in Figure 1D. The authors only mentioned these two ions and two water molecules added to specific position S1, S2, S3 and S4 line 528,529. Are these water molecules involved in the binding as water bridge? The authors should add S positions on the full structure of TASK3.
à Thanks for your comment. We describe point by point what we did according to your suggestion:
1) Now the structures of the studied ligands appear earlier in the paper, together with the previous figure 3. They cannot appear earlier than this Figure because the sequence of the results is:
1.1) Results from the exploration of PK-THPP binding site in TASK-3 by alanine mutagenesis
1.2) TASK-3 modeling and structural characterization screening
1.3) Structural-activity relationship (SAR) of THPP series against TASK-3 channel
They just appear now in the main figure of this section
2) Regarding to the two K+ ions and to add the S positions on the full structure of TASK3, they appear now in the first figure well described. Waters from the selectivity filter are not involved in the water bridge binding. Then we comment in line 491: `Water molecules that establish the water bridges are placed on the central cavity´
The authors should emphasize the results from the in vitro data. The location of the mutated amino acids should be improved (Figure 1D). The authors should also comment on the gain of function. Does the mutation P119A increased the receptor inhibition compared to WT?
à Thanks for your comments. We improved Figure 1D (now is Figure 2D) and to clarify the gain of function, we modify the legend of figure 2C. The phrase: "Significance was assessed using two- tailed Student’s t test. Asterisks indicate significance: *, p < 0.05; **, p < 0.01; ***, p < 0.001" was replaced by "Significance of the differences between the mutants and the WT result was assessed using two- tailed Student’s t test. Asterisks indicate significance: *, p < 0.05; **, p < 0.01; ***, p < 0.001. Other differences did not reach statistical significance."
Although it is tempting to speculate that P119A might be a gain of function mutation, we have refrained as the difference in the degree of inhibition with that of the wild type is not statistically significant.
Technical limitation
The rmsd from the 25 first ns is calculated on a constraint structure (1 kcal.mol.A-2 , line 542). Thus, it is obvious that the structure should remain close to its the initial conformation. The authors have taken 10 structures from this MD to consider the flexibility of the receptors. A constraint 25ns MD simulation cannot achieve a ’wide conformational sampling’ (line 444). Why not using a flexible docking procedure instead?
à Thanks for your comment. We indeed used a constrained molecular dynamic simulation to sample the possible conformations of the binding pockets in TASK3 channel. However, this protocol includes the flexibility of the amino acids side chains. The constraint is applied only to the backbone atoms that establish H-bonds form the alpha-helices. With this energetic restriction, the protein is free to “move” but always conserving the secondary structure. This protocol was previously used with positive results (See REF 13), allowing to explore several ligand-channel conformations using MDs combined with massive docking simulations and binding free energy calculations. A conventional flexible docking protocol does not allow to explore multiple ligand-receptor conformations. In addition, the MM-GBSA free energy methodology included in this work allows us to re-score poses from docking simulations to obtain better results (in score terms) than those that might be obtained by the use of a simple docking algorithm.
The authors have made three models of TASK3. They should take more care for describing the difference between the three structures. The Hole plots are not clear. The authors must specify what are the meaning of z and x axis. Moreover, they should add numbers to quantify the difference in pore size between the three models.
à Thanks for your comment. We answer point by point this comment:
1) In the Material and Methods -> TASK-3 modeling section, we describe the differences between the three TASK-3 models built in this work (lines 545 to 559):
`Since the three-dimensional (3D) structure of TASK-3 has not been solved, the sequence of human TASK-3 was downloaded (UniProtKB accession number: Q9NPC2) and three homology models were built using the following crystal structures as templates: TREK-2 (PDB: 4BW5), TREK-1 (PDB: 4TWK) and TWIK-1 (PDB: 3UKM). These structures have differences in the fenestration states (they could be open or closed); therefore, the different TASK-3 models were used to study the interactions between compounds of THPP series and TASK-3 with diverse fenestrations characteristics. The TASK-3 homology models were built according to the multiple sequence alignment published by Brohawn et al. [5] and optimized using Prime software [31,32]. The models (PDB files provided in the Supplementary Materials) were named according to the template and the fenestration state as follows: T3tre2OO (TASK-3 built from TREK-2 in Open-Open fenestration state), T3tre1CC (TASK-3 built from TREK-1 in Close-Close fenestration state), and T3twiOO (TASK-3 built from TWIK-1 in Open-Open fenestration state).´
2) Regarding the HOLE plots, these plots are used to measure the cavity radius in channels (See REF 38). In this work we studied two cavities, the central cavity or ion channel pore (Figure 3A), and the side-fenestrations (Figure 3B). In these plots are represented the radius of the cavity and the position (in z or x axis) where the radius was determined. Z axis is lined by the ion-conducting pore and now is described in figure 3 and X-axis is perpendicular to the ion-conducting pore and is lined by the access to the hydrophobic core of the lipid bilayer from the pore central cavity. X axis is also described now in figure 3.
3) Regarding to the pore size, now it is included in Figure 3 the central cavity bottlenecks of the models with open fenestrations compared to the pore size at that point of T3tre1CC to quantify the difference in pore size between the three models (about 8 Å). Also, the fenestration bottlenecks are included.
The authors have decided to correlate IC50 to their theoretical binding affinity. While the correlation is good and allow to choose among the different models of TASK3, the EQ4 appears to be false. The authors should be careful when comparing binding affinity to IC50 (line220). IC50 and affinity are not always equal (see the paper from Yung-ChiChengWilliam H.Prusoff: Relationship between the inhibition constant (KI) and the concentration of inhibitor which causes 50 per cent inhibition (I50) of an enzymatic reaction)
à Thanks for your comment. We answer point by point this comment:
1) In fact, we haven´t use EQ4 but R2. We removed EQ4 and re-write the text according to your suggestions. Now is written (lines 197 to 200):
29 compounds of THPP series with IC50 < 100 µM (supplemental Table S1) were docked against the TASK-3 models at the last frame of the 25ns MDs, then the MM-GBSA relative ΔGbind energy was calculated to correlate (Eq. 4) the THPP series structural binding mode and their reported activity against TASK3 [14].
In the Methods section is written (Line 620 to 623): The computed relative binding free energies were plotted correlated against experimental IC50 values for the 29 compounds of the THPP series selected in this study. The degree of statistical correlation (R2) between the both experimental IC50 and computed relative ΔGBind values is reported.
2) The paper from Cheng and Prusoff (1973) was checked. They say “The analysis described shows KI does not equal I50 when competitive inhibition kinetics apply; however, KI is equal to I50 under conditions of either noncompetitive or uncompetitive kinetics”.
We consider that an ion channel blocker could be in the last two cases. Ion channels are not enzymes and the binding affinity of an inhibitor (or an activator) to an ion channel is calculated from the concentration response data (doi: 10.3389/fphar.2018.00150).
Minor point.
Table 1 does not bring any input for the paper. It should be provided as supplementary material.
à Thanks. Now it is table S1.
The author should use the term ΔMM-GBSA instead of ΔGbind while they are not calculating any entropy penalty.
à We use indistinctly MM-GBSA ΔGbind or relative ΔGbind (line 596) within the text considering that entropic changes are not applied.
Line 133 ’25 ns MDs to stabilize them’. MD simulation do not stabilize a molecule. A MD simulation samples the conformational space of the system.
à We thank the reviewer for this observation. Now is written only `The TASK-3 models were subjected to 25 ns molecular dynamics simulation (MDs) to stabilize them. ´
The authors should be more precise on the protein RMSD calculation by providing the atoms they were choosing to calculate it.
à The information regarding the atoms chosen to calculate the RMSD is the figure legends (Now Figure S1 and S6). In the protein, the RMSD was calculated from the backbone atoms and in the ligand PK-THPP is clarified now that it was calculated from the heavy atoms (Figure S6).
Left panel from figure 5 is useless for the paper. Instead of a color code, the authors should label the amino acids on the right panel of figure 5.
à Thanks for your suggestions. We describe point by point what we did according to it:
1) We consider that figure 5 (left panel) is important for the paper message, although not crucial. Now this figure is in the Supplementary Material because it shows that PK-THPP poses docked in T3tre2OO model exhibit lower RMSDs than in the other two models.
2) We rather to use a color code because the binding site is integrating by 10 amino acids, and to label all the residues in the figures add noise to the artwork.
The paragraph between line 425 and 438 seems to be out of the scope of this paper as it is a comment on a previous paper (REF14).
à No, it is not. In this paragraph, we are discussing the results from Coburn et al. (2015), on the light of our results. Consider that Coburn et al. (2015) did not perform site directed mutagenesis analysis, only the compound synthesis and to measure their activities.
The pharmacophore study is bringing no new insight for this study as the ‘compounds derived from the same scaffold’ (line 587 588). The authors are solely focusing on the lack of H-bond acceptor on the compound 17 which can be directly observed on the molecular structure.
à Considering the pharmacophore definition “an ensemble of steric and electronic features that is necessary to ensure the optimal supramolecular interactions with a specific biologic target and to trigger (or block) its biologic response” (https://doi.org/10.1016/B978-1-4377-1679-5.00001-6) is easier for the reader (mainly for biologists) to understand the molecules with these features than only by their chemical structure. With this purpose was included the pharmacophore features and we believe it necessary for the comprehension of the paper message.
Line 624: 300°K should be 300 K.
à Thanks for the comment, we changed the text according to the suggestion.

Reviewer 2 Report
The present manuscript of Ramirez et al can be accepted to publication, after some minor modifications will be performed:
-row 55: degree sign should be without the horizontal line
-row 57: Prozac is a trade name and should be written with the distinctive sign
-row 106: "...residues ( L12..." should be written without space before "L"
-row 216: font inside the table should be uniformized
-row 213: in the hydrophobic column, the content should be presented otherwise, the content isn't understandable
-row 302: font inside the table should be uniformized
-row 375: correct is "X-ray"
-row 378: "...like Br fluoxetine and norfluoxetine...". I think that a comma is missing
-row 450-453: (1), (2), (3) and (4) have a strange formatation
-row 490-492: concentration of each component should be presented like "NaCl 82.5 mM"
-row 504: "A/D" should be written as "analog to digital"
-row 575-595: all text from equations should have the same font
-row 624: the degree sign is not needed in Kelvin temperatures
-row 633: "...). The..." one out of the two spaces should be removed
Author Response
Response to Reviewer 2 Comments
Comments and Suggestions for Authors
The present manuscript of Ramirez et al can be accepted to publication, after some minor modifications will be performed:
-row 55: degree sign should be without the horizontal line
-row 57: Prozac is a trade name and should be written with the distinctive sign
-row 106: "...residues ( L12..." should be written without space before "L"
-row 216: font inside the table should be uniformized
à We thank the reviewer for these observations. All suggestions were addressed in the text.
-row 213: in the hydrophobic column, the content should be presented otherwise, the content isn't understandable
Did you refer to row 231 instead of 213? If it is, you can visualize the content on Table S4. Here it is a small space to show it. You can check for example, on table S4, the compound 17b and to contrast it with table 2. Compound 17b has 5 hydrophobic interactions which are the interactions # 1, 2, 4, 7, 8 of table S4. # 1, 2 and
4 are through the biphenyl group of compound 17b, #7 through the pyrimidine group and # 8 through the substituent
-row 302: font inside the table should be uniformized
-row 375: correct is "X-ray"
à We thank the reviewer for these two observations. They were addressed in the text.
-row 378: "...like Br fluoxetine and norfluoxetine...". I think that a comma is missing
à Now the sentence is (Line 395) “But in the `down state´, the side chain of L320 is in the side-fenestration providing, with other residues such as F316, a hydrophobic environment close to the selectivity filter in which drugs like Br fluoxetine and norfluoxetine binds”.
-row 450-453: (1), (2), (3) and (4) have a strange formatation
à They are in cursive letter to highlight the points
-row 490-492: concentration of each component should be presented like "NaCl 82.5 mM"
-row 504: "A/D" should be written as "analog to digital"
à We thank the reviewer for these two observations. They were addressed in the text.
-row 575-595: all text from equations should have the same font
à All of them are in “Cambria Math 10”
-row 624: the degree sign is not needed in Kelvin temperatures
-row 633: "...). The..." one out of the two spaces should be removed
à We thank the reviewer for these two observations. They were addressed in the text.

Reviewer 3 Report
Manuscript submitted by Ramírez et. al "Structure/activity analysis of TASK-3 channel antagonists based on a 5,6,7,8 tetrahydropyrido[4,3-d]pyrimidine" in Int. J. Mol. Sci.. It requires revision to improve result description and references.1. In introduction section and section 2.2 binding thermodynamics part cite article: Mol Pharm. 2017 1;14(5):1656-1665. Mol Pharm. 2018, 2;15(4):1445-1456.
Author Response
Response to Reviewer 3 Comments
Manuscript submitted by Ramírez et. al "Structure/activity analysis of TASK-3 channel antagonists based on a 5,6,7,8 tetrahydropyrido[4,3-d]pyrimidine" in Int. J. Mol. Sci.. It requires revision to improve result description and references.
1. In introduction section and section 2.2 binding thermodynamics part cite article: Mol Pharm. 2017 1;14(5):1656-1665. Mol Pharm. 2018, 2;15(4):1445-1456.
à We thank the reviewer for this observation. The references were included in the introduction section.

Round 2
Reviewer 1 Report
The authors have largely improved their manuscript. The figures are now relevant and support the findings.
However, my main concern about the importance of T93 and specially its involvement in a hydrogen bond has still not been addressed. The authors should remove any of their speculations about this hydrogen bond since it is not in line with the experimental results.
Technical concerns
2) Regarding to the two K+ ions and to add the S positions on the full structure of TASK3, they appear now in the first figure well described. Waters from the selectivity filter are not involved in the water bridge binding. Then we comment in line 491: `Water molecules that establish the water bridges are placed on the central cavity´
From where this water molecule comes from? Was it placed manually by the authors prior to the MD?
Author Response
Comments and Suggestions for Authors The authors have largely improved their manuscript. The figures are now relevant and support the findings. However, my main concern about the importance of T93 and specially its involvement in a hydrogen bond has still not been addressed. The authors should remove any of their speculations about this hydrogen bond since it is not in line with the experimental results. -->Thank you for this constructive comment. For the previous revision, we tried to increase the number of experiments for this particular mutant (previous n number = 3), as our modeling data points out an important role of T93. Unfortunately, we were not able to deliver this data for the previous deadline. However, in the meantime we acquired more data and doubled the number of the experiments. Please find the updated Figure 2C, where the average block of T93A mutant decreased from 79.7% to 65.6%, and is significantly different from the wild type (p = 0.022). In light of this new evidence, we strongly think that this data should remain in the main manuscript. Now you can read in the legend of figure 2 (lines 146 to 150): Residues identified as THPP binding site (`hits’) have significant levels of p < 0.01 and p < 0.001 except T93 (p < 0.05), important from the modelling point of view. D. Residues of PK-THPP binding site are facing the pore and the fenestrations as it is shown in TASK-3 models in the down (T3tre2OO) and up (T3tre1CC) states (lateral view from the pore). Hits (Chain A: T93, L122 and Q126; Chain B: G231, G236, A237, L247 and T248) are represented in licorice with individual colors. Also, a change was done in the results 2.5 section “Interaction of PK-THPP with residues of TASK-3 binding site by molecular dynamics simulations (MDs)”. From line 345 to 348: We found that the drug interacts during the entire MDs with residues L122 and L239 in both chains (Figure 6B), and with other residues with less frequency or not in both chains such as T93, Q126, G236, L244, L247 and T248. Except T93, the other residues were identified by alanine scanning as highly significant for the binding (Figure 2C). Technical concerns 2) Regarding to the two K+ ions and to add the S positions on the full structure of TASK3, they appear now in the first figure well described. Waters from the selectivity filter are not involved in the water bridge binding. Then we comment in line 491: `Water molecules that establish the water bridges are placed on the central cavity´ From where this water molecule comes from? Was it placed manually by the authors prior to the MD? -->There is not only one water molecule to establish the water bridge between Q126 and PK-THPP. But all of them have the same origin and they were not placed manually. All water molecules in the system (in total 11466 molecules), except those two at the selectivity filter, were placed using the Desmond system builder tool. This tool uses an algorithm that randomly solvate the TASK-3 channel embedded in the lipid membrane with pre-equilibrated SPC (Single Point Charged) water molecules. As an example, it is shown in the next figures the behavior of one of the water molecules that establish the water bridge between Q126 and PK-THPP during 11.76 ns. Fig 1 (See the attached Word document). Different position of one of the water molecules that establish a water bridge between Q126 and PK-THPP (green) from the beginning of the 250 ns MDs (0 ns) until 27 ns, when the water bridge is established. For simplicity, only the protein (one subunit red and the other white) and the waters are shown. Membrane was omitted. In Fig 1 you can appreciated one of the water molecules that establish a water bridge between Q126 and PK-THPP. Initially it is in the extracellular space (0ns), then in the intracellular space (20ns) and finally at the central cavity (27ns). The molecule movement from the extracellular to the intracellular space occurs through the periodic box (the system is under periodic boundary conditions). This set of boundary conditions are chosen to approximate our system to a large (infinite) system by using a unit cell (purple box in Fig 1). When the distance between the oxygen atom from the water molecule highlighted in Fig 1 and the hydrogen atom from Q126 is measured, we can observe that this water molecule moves in the water bulk along the first 26 ns of the simulation, then this water stablish the water bridge interaction with Q126 for 4.6 ns, and then moves in the central cavity close the Q126 residue. Later, between 50-100 ns, two other water bridge interaction involving this water can be detected. These interactions are of 2.5 ns and 4.6 ns. Fig 2. (See the attached Word document) Distance from the oxygen atom of the water molecule highlighted in Fig 1, and the hydrogen atom from Q126 during the 250 ns MDs. This water molecular cover about 10% of the water bridge interaction between Q126 and PK-THPP. Certainly, other water molecules will cover the rest (about 138 ns) of the observed water bridge interaction; but all of them has the same origin: they were placed using the Desmond system builder tool and not manually.
